# Highlights of Major Progress on Single-Atom Catalysis in 2017

**Yalin Guo [1,2] , Rui Lang [1] and Botao Qiao [1,]***

[1]     Dalian Institute of Chemical Physics, Chinese Academy of Sciences, Dalian 116023, China;
        guoyalin@dicp.ac.cn (Y.G.); langrui@dicp.ac.cn (R.L.)
[2]     University of Chinese Academy of Sciences, Beijing 100049, China
[*]     Correspondence: bqiao@dicp.ac.cn

**Abstract:** Single-atom catalysis has rapidly progressed during the last few years. In 2017, single-atom catalysts (SACs) were fabricated with higher metal loadings and designed into more delicate structures. SACs also found wide applications in C1 chemical conversion, such as selective oxidation of methane and conversion of carbon dioxide. Both experimental characterizations and computational modeling revealed the presence of tunable interactions between single atom species and their surrounding chemical environment, and thus SACs may be more effective and more stable than their nanoparticle counterparts. In this mini-review, we summarize the major achievements of SACs into three main aspects: (a) the advanced synthetic methodologies, (b) catalytic performance in C1 chemistry, and (c) strong metal-support interaction induced unexpected durability. These accomplishments will shed new light on the recognition of single-atom catalysis and encourage more efforts to explore potential applications of SACs.

**Keywords:** single-atom catalysis; catalyst fabrication; C1 chemical conversion; strong metal-support interaction; reaction mechanism

---

## 1. Introduction

Single-atom catalysis, which was proposed a few years ago [1], has now become a new frontier in heterogeneous catalysis [2]. Compared with supported nanoparticle (NP) catalysts that consist of various kinds of active sites (corners, defects, etc.) [3], "single-atom catalysts" (SACs), containing only individual metal atoms on the solid support, can provide uniform, structurally well-defined active sites [4]. In recent years single-atom catalysis has attracted considerable attention and made very rapid progress [5–9]. Wang et al. summarized recent experimental and computational reports to illuminate the bonding in SACs and its relationship with catalytic performance [10]. During 2017, Liang et al. moved and controlled the atoms on the surface with the aid of a scanning tunneling microscope (STM), helping to understand catalysis at single-atom level [11]. In addition, various other remarkable accomplishments have come to light in 2017. First, synthetic approaches became more rational to tune the electronic property of active metal species and construct ordered morphology of support. Second, SACs found more applications in C1 chemical (methane, methanol, $CO_2$, and CO) conversion. Moreover, SACs exhibited unexpected durability compared with the NP counterparts, which can be attributed to the unusual strong interaction between metal atoms and supports. Last year, we summarized the major progress in 2015 and 2016 [12]. In this review, we would like to talk about the above respects in detail for the convenience of readers in order for them to know about the most recent progress.

## 2. The Preparation Strategies of Single-Atom Catalysts (SACs)

The fabrication of stable SACs has, at least at the present state, been considered as a great challenge since single-atom catalysts are believed to be somewhat thermodynamically unstable. However, some cases have proved that this may not be true. For example, Au atoms were found to form strong covalent metal-support interaction with $FeO_x$ which showed superior reaction durability than their NP counterparts [13]. Others reported that heating Pt NP with $CeO_2$ support at high temperature in the presence of oxygen can generate Pt SACs [14,15]. Therefore, appropriate preparation methods to synthesize stable SACs with unique structures are summarized below.

### 2.1. SACs with Ordered Structure

Thermal treatment, usually under an inert atmosphere, of carbon-based materials with uniform pore sizes and controllable shapes has been gradually developed as a reliable strategy to prepare SACs with the desired morphology. Li et al. anchored Fe atoms on the inner wall of hollow *N*-doped carbon (CN) tubes by sacrificing nanorod template [16], as shown in Figure 1. The synthetic procedure included three steps: (1) coating ferric oxide nanorod with organic polymers; (2) carbonizing at high temperature; and (3) etching the metal oxide with acid. The Fe SAC showed excellent activity toward the hydroxylation of benzene with 45% conversion and 94% phenol selectivity. In addition, this methodology can be applied to a group of transition-metal (M) single atoms (SA) dispersed on CN materials, the so-called SA-M/CN (M = Fe, Co, Ni, Mn, FeCo, FeNi), by varing metal precursors or polymers.

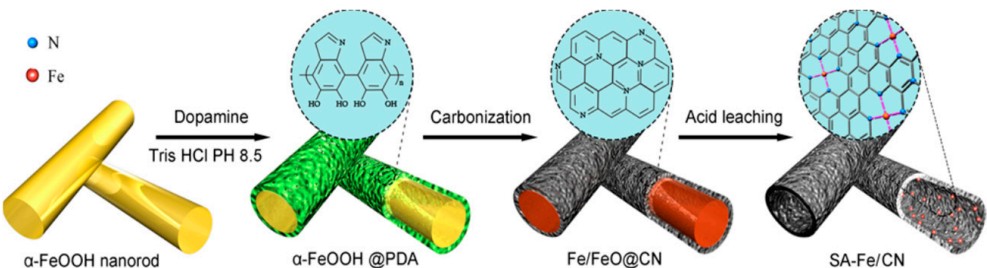

**Figure 1.** Synthetic procedure for SA-Fe/CN. Reprinted with permission from Ref [16]. Copyright 2017, American Chemical Society.

Metal organic frameworks (MOFs), bridging metal atoms with ligands to form high surface area, rich micropore, three-dimensional ordered crystal frameworks, have been demonstrated as promising precursors to obtain SACs [17]. Wu et al. prepared a MOF-derived single atom Fe catalyst by directly bonding Fe ions to imidazolate ligands [18]. The well-dispersed $FeN_4$ active sites are embedded into porous carbon to prevent agglomeration in an oxygen reduction reaction (ORR). Increasing thermal-treat temperature leads to more active sites and enhanced ORR activity.

Ru coordinates with the skeletons of UiO-66 MOFs can generate good hydrogenation SACs [19]. In Figure 2, Cao et al. prepared a $Ru/ZrO_2@C$ SAC with 0.85 wt% Ru loading and compared with commercial Ru/C in converting levulinic acid (LA) to γ-valerolactone (GVL). Full conversion of LA and quantitative yield of GVL are achieved in both $Ru/ZrO_2@C$ and Ru/C catalyst. However, no matter whether in water or in high protic aqueous solution, $Ru/ZrO_2@C$ displayed almost the same catalytic performance upon multiple recycling, suggesting Ru atoms were highly dispersed on nanotetragonal $ZrO_2$, and embedded in the amorphous carbon. Temperature programmed reduction (TPR) results showed that the strong metal–support interaction between Ru and $ZrO_2$ may be in favor of this excellent stability [20].

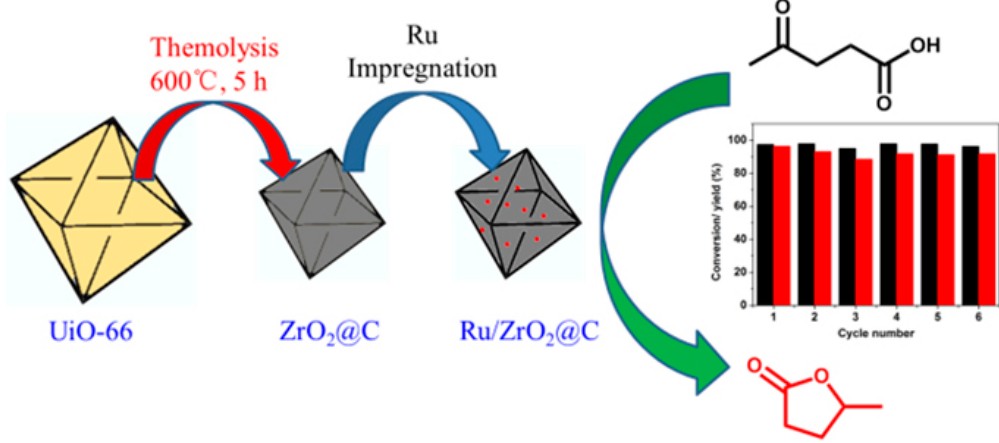

**Figure 2.** Synthesis and durability of metal organic framework (MOF)-derived Ru single-atom catalyst (SAC). Reprinted with permission from Ref [20]. Copyright 2017, The Royal Society of Chemistry.

## 2.2. Single-Atom Alloy (SAA)

Single-atom alloy (SAA) presents one type of important SACs, as suggested in previous work and review [12,21]. Pei et al. extended the scope of Pd-IB Metal SAA system and chose Cu as the partner of Pd [22,23]. The Cu-alloyed Pd SAA ($CuPd_{0.006}/SiO_2$) showed ~85% selectivity at 100% conversion for the selective hydrogenation of acetylene to ethylene under a simulated front-end hydrogenation process in industry. However, direct imaging of isolated Pd on Pd-Cu SAA is not an easy task due to the close Z contrast between these two elements, so Extended X-ray absorption fine structure (EXAFS) and density functional theory (DFT) calculations were used to identify the neighboring environment of Pd atoms. Besides, the surface property of Pd/Cu SAA can be investigated by polarization-dependent reflection absorption infrared spectroscopy (PD-RAIRS) using CO as a probe molecule [24]. The surface Pd coverage can even be quantified from the CO peak area of PD-RAIRS and Auger electron spectra (AES). However, the calculated coverage is different because the detected electrons with AES come from both surface and subsurface Pd atoms, while RAIRS only detected CO bounding to Pd atoms at the immediate surface.

Chen et al. dispersed Pt atoms on the surface of Ni particles forming Pt/Ni SAA by the galvanic reduction method. Pt/Ni SAA exhibited relatively high activity for the hydrolytic dehydrogenation of ammonia−borane, due to the synergistic effect between Pt and Ni [25]. A kind of stable molten metal alloy catalyst for the pyrolysis of methane into hydrogen and carbon was synthesized by dissolving active metals (Ni, Pt, Pd) in inactive low–melting temperature metals (In, Ga, Sn, Pb) [26]. During the reaction, the insoluble carbon floats to the surface where it can be skimmed off, making the catalysts anti-coking. The molten alloys are called "liquid SAA" for the active metals are atomically dispersed.

## 2.3. Support Effect

Apart from changing the support structure or doping a second metal to form SAAs, the surface properties, e.g., defects, components, or impurities, of support can also influence SACs' catalytic behavior. Yang et al. studied the support effects in ORR by separately depositing Pt atoms on TiN and TiC support with the same configuration. The $Pt_1/TiC$ catalyst showed higher activity and selectivity toward $H_2O_2$ via a 2 $e^-$ pathway, whereas the $Pt_1/TiN$ surface was poisoned by strong affinity to oxygen species. In this case, the importance of supports is proved since supports also participate in the surface reaction [27].

### 2.3.1. Functional Groups on The Support

Carbon-based materials usually utilize surface O, N, and S-containing functional groups to anchor metal sites. It is found that Pt/C interaction strengthened on the carbon support with higher oxygen

concentrations, which is more dependent on charge transfer rather than frontier–orbital hybridization. In general, the higher concentration of oxygen-containing groups (OCGs) is in favor of the stability and catalytic activity of the catalyst. DFT calculations suggested that tailoring the carbon support by OCGs or other light-element can provide a new route to improve the tolerance of Pt/C catalysts against CO poisoning [28] and agglomeration [29]. Actually, heat-treating cobalt salts and graphene oxide in ammonia atmosphere can generate cobalt SAC on nitrogen-doped graphene (Co-NG), with up to 7.94 at% nitrogen concentration. The Co-NG catalyst shows high activity and excellent stability for selective alcohol oxidation. Nitrogen promoted a metal-support interaction by electron transfer [30]. A synergetic effect between doped N and isolated Pt sites was also found for a carbon black-supported Pt single-atom electrocatalyst with CO/methanol tolerance for ORR. The strong interaction between Pt and N can even prohibit Pt oxidation in air. The acidic single-cell with such a catalyst as cathode exhibited power density up to 680 mW/cm$^2$ at 80 °C [31]. A high-density atomically dispersed Fe anchored on S-doped NC ORR catalyst (Fe/SNC) was synthesized [32]. The incorporated sulfur, emerging as a thiophene-like structure (C–S–C), plays an important role in reducing the electron localization around Fe centers, and facilitates the complete 4 e$^-$ ORR in acidic media. The above examples demonstrated the outstanding characteristics of heteroatom (O, N, S) doped carbon supports as ideal carriers for highly dispersed metal centers.

### 2.3.2. Surface Defects of Support

Surface defects have been intensively investigated on two-dimensional materials such as boron nitride (BN) and MoS$_2$. Chen et al. investigated transition metals (Mo, Ru, Rh, Pd, and Ag) singly dispersed on the defective BN monolayer with a boron monovacancy as N$_2$ fixation electrocatalysts by DFT computations. Results suggested that Mo single atom supported by a defective BN nanosheet may be highly active for N$_2$ fixation at room temperature [33]. Isolated Co atoms are covalently bonded to sulfur vacancies on MoS$_2$ monolayer sheets forming Co–S–Mo interfacial sites. This Co SAC can reduce the hydrodeoxygenation reaction temperature from the typically 300 °C to 180 °C [34].

### *2.4. External Forces Induced SAC Synthesis*

### 2.4.1. Iced Photochemical Reduction

Apart from the above thermal-treatment, ultraviolet (UV) irradiation of frozen H$_2$PtCl$_6$ aqueous solution also can generate atomically dispersed Pt stabilizing on various carbon-based or metal-oxide substrates. The ice lattice confined Pt precursor migration and prevented the nucleus formation of photochemical reduction products Wang et al. compared traditional UV irradiation with UV irradiation accompanying iced-photochemical process. Obviously, the traditional UV irradiation of H$_2$PtCl$_6$ aqueous solutions (as shown in the upper line) produced Pt nanocrystals formed by the agglomeration and nucleation of Pt atoms. However, Pt single atoms dispersed in ice could be attained by exposing the frozen solution of H$_2$PtCl$_6$ to UV irradiation with a low temperature to reduce Pt$^{4+}$. The Pt$_1$/mesoporous carbon material prepared following the new UV irradiation process is an effective electrocatalyst for the hydrogen evolution reaction (HER) with an overpotential of only 65 mV at a current density of 100 mA/cm$^2$, superior to state-of-the-art platinum/carbon. This iced-photochemical reduction can be extended to gold and silver [35].

### 2.4.2. Electrodeposition

Electrons were also used as driving force for active metal deposition. Bard et al. used femtomolar concentrations of PtCl$_6$$^{2-}$ to limit the Pt atoms plated on the electrode. Isolated single Pt atoms to 9-atom small clusters were supported on an inert bismuth ultramicroelectrode [36]. Taking the potential at a certain current density as a measure of the relative rate of the HER, they found that the potential shifted positively as the Pt cluster size increased, thus single atoms showed a larger overpotential than bulk Pt. A similar potential-cycling method was used to synthesize Pt SAC on

CoP-based nanotube [37], which exhibited HER activity comparable to commercial Pt/C in neutral media. The Pt mass activity of SAC is 4 times of that of Pt/C, and the stability is also much better. These results provide an opportunity to fabricate catalytic structures on an atom-by-atom basis and directly evaluate catalytic activity after deposition.

## 3. Novel Applications of SACs

Compounds containing only one carbon atom, such as methane, carbon monoxide, carbon dioxide, and methanol, are classified as C1 chemicals. Direct conversion of simple C1 compounds into high-valued chemicals can be counted as one of the major challenges in chemical engineering. In 2017, single-atom catalysis was applied as an advanced strategy in this research field. For example, Rh SACs were used to activate methane, the most inert C1 chemical, forming methanol and acetic acid. Oxygen-containing C1 chemicals, such as formic acid can be reformed to generate $H_2$ by the aid of Pt SACs. Moreover, SACs were even capable of completing the traditional homogeneous catalyzed process with satisfying performance. Once again, single-atom catalysis was demonstrated to be the bridge between homogeneous and heterogeneous catalysis [38,39].

### 3.1. $CH_4$ Conversion

Methane is the main constituent of natural gas and biogas. Recently it has also been found in "flammable ice" at the continental slopes of many oceans including the South China Sea. Methane, in view of an abundant new source of energy with vast reserves and resources, has been regarded as a secure supply of chemicals and fuels in the future. However, as the most inert C1 chemical, activation of methane is a challenging process of great importance in modern catalysis [40]. Flytzani-Stephanopoulos et al. fabricated a kind of isolated rhodium catalysts anchored on a zeolite or $TiO_2$ support, which can catalyze the direct conversion of methane to methanol or acetic acid in aqueous solution [41]. They prepared the catalysts using simple heat-treating protocol on zeolite (ZSM-5) and ultraviolet irradiation on $TiO_2$. The acidity of the support can change the final product (methanol or acetic acid). Transformation from methane to oxygenates needs a two-step pathway. First, activation of methane occurs on the isolated $Rh^+$ species forming $Rh–CH_3$ in the presence of $O_2$, which then is transformed via two routes: O insertion to generate methanol or CO insertion to generate acetic acid. Following the hydrolysis step, single $Rh^+$ can take part in the next catalytic cycle. Theoretically, Rh atoms dispersed on $ZrO_2$ surface ($Rh_1/ZrO_2$) can also activate methane to energetically stabilize $-CH_3$ intermediates. After optimizing conditions, $Rh_1/ZrO_2$ selectively oxidized methane to methanol in $H_2O_2$ aqueous solution, circumventing the complete oxidation to $CO_2$ with Rh NPs [42]. These works demonstrated that selective and direct conversion of methane to specific oxygenates can occur on the atomic metal sites and provided new insight into the development of heterogeneous catalyzed industrial processes.

### 3.2. Methanol and Formic Acid Reforming

Two supported Pt SACs were used in the reforming reaction of methanol or formic acid to generate $H_2$, a clean energy. Generally speaking, the dehydrogenation reaction always competes with dehydration reaction which forms CO and $H_2O$. Pt and Pd catalysts are active for both dehydrogenation and dehydration, while Cu surfaces are highly selective toward dehydrogenation. Sykes et al. substituted single Pt atoms into the Cu lattice and found the dehydrogenation of formic acid to $CO_2$ and $H_2$ was highly selective [43]. Pt atoms dispersed over $\alpha$-MoC showed outstanding hydrogen production activity in the low-temperature aqueous-phase reforming of methanol (APRM) process [44]. When Pt loading decreased to 0.2%, the atomically dispersed Pt became the dominant species on $\alpha$-MoC after high-temperature activation process, generating an exceptionally high density of electron-deficient $Pt_1$ sites for the adsorption/activation of methanol. Meanwhile, $\alpha$-MoC shows high water-dissociation activity, producing abundant surface hydroxyls to accelerate the reformation of intermediates at the interface between Pt and $\alpha$-MoC. Both effects make Pt/$\alpha$-MoC as a catalyst with excellent efficiency and good stability in the base-free APRM process at working temperature of 150–190 °C.

### 3.3. CO₂ Conversion

The vast majority of anthropogenic emission of $CO_2$, the most notorious greenhouse gas, comes from combustion of fossil fuels. If $CO_2$ emission continues at the present rate, the Earth's surface temperature would probably exceed historical values as early as 2047, which will definitely harm the worldwide ecosystems and biodiversity [45]. DFT studies suggested the possibilities of converting $CO_2$ to other chemicals by SACs [46,47]. Nanoscaled Cu, although it always suffers from oxidation, has previously been identified as an active metal for $CO_2$ conversion. Cu atoms confined in Pd lattice exhibit enhancement for $CO_2$-to-$CH_4$ conversion by two major effects: (1) providing the paired Cu−Pd sites for the enhanced $CO_2$ adsorption and the suppressed $H_2$ evolution; (2) elevating the d-band center of Cu sites for the improved $CO_2$ activation. Consequently, $Pd_7Cu_1$−$TiO_2$ achieved the highest photo-catalytic performance with 95.9% selectivity [48]. In electroreduction of $CO_2$ to CO, a high loading Ni SAC reached a TOF value of 5273 $h^{-1}$, with a Faradaic efficiency over 71.9% at an overpotential of 0.89 V and a current density of 10.48 mA/$cm^2$ [49].

Layered double hydroxides (LDHs) are superior base catalysts, because the $OH^-$ groups are 6-fold coordinated with divalent and trivalent cations. An isolated Ru catalyst was synthetic on the surface of LDH and utilized to hydrogenate $CO_2$ into formic acid in a basic medium [50]. The isolated Ru species with rich electron were generated with the help of strong Bronsted $OH^-$ ligands (Figure 3), which are in favor of $CO_2$ adsorption near the active Ru center. The Ru/LDH SAC obtained comparable or higher turnover number (TON) and turnover frequency (TOF) values than any other reported heterogeneous catalyst systems. The catalytic cycle for $CO_2$ hydrogenation was proposed as below.

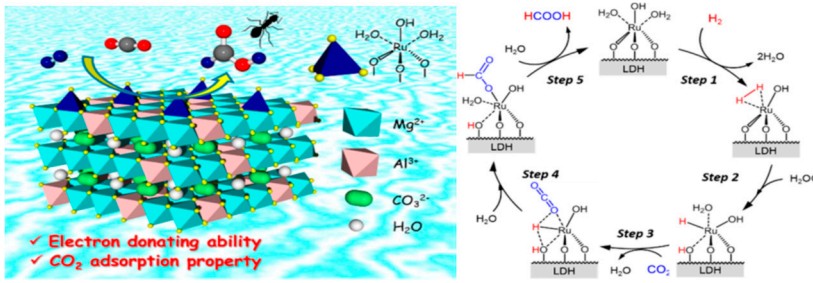

**Figure 3.** Ru/LDH SAC catalyzed $CO_2$ hydrogenation to formic acid. Reprinted with permission from Ref [50]. Copyright 2017, American Chemical Society.

### 3.4. CO Conversion

CO oxidation and water gas shift reactions (WGSR) have been extensively investigated in heterogeneous catalysis due to their importance in both industrial applications and fundamental studies. Some reviews about single-atom catalysts in this field have provided deep insight [2,9,51,52]. In 2017, both DFT studies [53] and experimental research had continuous progress in WGSR reaction. Guan et al. developed two SACs for WGSR. In $Rh_1$/$TiO_2$ system, Rh single atoms promoted the formation of oxygen vacancies on the $TiO_2$ support and prohibited the $H_2$ dissociation, leading to an overall 95% CO conversion with no methanation at 300 °C, even under $CO_2$- and $H_2$-rich stream [54]. Similarly, Pd/$FeO_x$ SAC also facilitated oxygen vacancies formation and proceeded through a redox mechanism with low activation energy [55].

CO oxidation is one of the most representative probe reactions in single-atom catalysis, and both activity and mechanism were studied in isolated Pt and Pd on various supports [56], e.g., $CeO_2$ [57–59], $Al_2O_3$ [60,61], and MgO [62]. Lou et al. systematically investigated the support effects for the CO oxidation reaction catalyzed by isolated Pt atoms on metal oxides with different oxidation-reduction potential: $Fe_2O_3$, ZnO, γ-$Al_2O_3$. They revealed that the catalytic performance of Pt SACs is affected mainly by the support properties. Both −OH groups on support surfaces and the added $H_2O$ improve the activity of these three SACs significantly [63]. The same group also observed the movement of Pt atoms in high-loading Pt/$Fe_2O_3$ SAC at 250 °C under different gas environments and found: (1) $O_2$ did

not sinter the Pt single atoms, (2) both CO and $H_2$ facilitated the movement of the Pt atoms, especially in the presence of $H_2O$. Therefore, interaction between metal species and the support needs to be strong enough to resist the possible sintering under working conditions [64]. Christopher et al. demonstrated the isolated Pt on certain reducible support ($TiO_2$) provides the most efficient metal utilization [65]. Although there have been many reports about the successful preparation of SACs and their supreme performance, we should still mention that in some cases, metal clusters/NPs may exhibit higher activity than their SAC counterparts. For example, $FeO_x$ supported Ir sub-nanometer clusters exhibited higher activity for CO oxidation with or without the presence of $H_2$ than Ir atoms [66].

### 3.5. Photoelectrocatalytic Reactions

Currently, electrocatalysts are playing a more and more vital role in the continuing development of electrochemical storage and conversion devices. However, highly efficient electrocatalysts need more rational control of size, shape, composition and structure. By the aid of rapidly emerging studies on SACs, novel Pt-group and non-noble metal electrocatalysts have been applied in oxygen reduction reaction [67,68], oxygen evolution reaction [69], hydrogen evolution reaction [67,70], and other related reactions. The unique single-atom dependent performance in electrocatalysis was summarized in a good recent review [71].

Photo-driven remediation of water contamination is a promising strategy for environmental protection. Ag SACs supported on carbon-based materials were independently reported by two groups [72,73]. One showed that combining singly-dispersed Ag and CQDs onto ultrathin $C_3N_4$ can form a novel ternary photocatalyst with enhanced photo-response, which offers broad-spectrum (from UV to near-infrared light) utilization of solar light for the degradation of naproxen [72]. The other group took advantage of the synergistic effect between single-atom Ag and $C_3N_4$ to prepare Ag functionalized mesoporous graphitic carbon nitride hybrid, which showed an excellent performance for the degradation of bisphenol A [73].

### 3.6. Heterogenization of Homogeneously Catalyzed Processes

Alongside the routine heterogeneous catalyzed abatement of environmental contaminants [74,75], SACs were reported to exhibit good performance in traditionally homogeneous catalytic processes, such as hydroformylation [38,39], hydrosilylation [76], C–H bond activation/oxidation [77] and C–C bond coupling [78,79]. SACs are, thus, predicted to be the bridge connecting homo- and heterogeneous catalysis, as summarized in a minireview [80]. Gold/carbon (Au/C) catalysts comprising single-site cationic Au entities were applied in acetylene hydrochlorination. In situ X-ray absorption fine structure study under reaction conditions showed that the high activity was correlated with the Au(I)/Au(III) ratio, so a mechanism based on a redox couple of Au(I)-Au(III) species was proposed by computational modeling [81].

## 4. Strong Metal-Support Interaction of SACs

The component and structure of active sites on SACs may be quite different from those on NPs, resulting the diverse catalytic activity and reaction mechanism. For example, DFT studies suggested a positively charged single Pt atom on $TiO_2$(110) may exhibit very high WGSR activity at low-temperature range, for $TiO_2$-supported Pt clusters and Pt atoms provided different active sites [82]. Li and coworkers successfully prepared catalyst of singly dispersed Rh atoms anchored on the inert $SiO_2$ support. The calculation results also confirmed that the Rh SAC and Rh NPs underwent different reaction mechanisms [83]. The unique interaction between isolated metal and support may be the origin for diverse catalytic performance. A quantitative profile for exploring metal–support interaction was provided by considering the highest occupied state in single-atom $Rh_1/VO_2$ catalyst [84], since for this Rh SAC, the impacts of size, shape, and orientation were negligible of the metal particles at the metal–support interface. At 341.0 K, $VO_2$ undergoes a phase transition between metal and insulator. The highest occupied state of Rh has a significant influence on determining the different apparent

activation energies of the two phases of $VO_2$. Thus, changing the highest occupied states may also be effective for other metals to tune catalytic properties in $NH_3BH_3$ hydrolysis reaction.

Owing to the strong metal–support interaction, SACs exhibited unexpected durability for electrocatalytic ORR in an acidic solution [85], and photocatalyzed water splitting [86]. $Al_2O_3$ is a common support for Pt in industrial and environmental applications. Yan et al. applied a modified sol-gel solvent vaporization self-assembly method to prepare an atomically dispersed Pt catalyst. Pt atoms strongly anchored in the inner surface of mesoporous $Al_2O_3$, staying in a four-oxygen coordination mode and being stabilized by coordinatively unsaturated pentahedral $Al^{3+}$ centers [87]. This Pt SAC maintained CO oxidation activity for one month. It was also highly stable in selective hydrogenation under extreme conditions, providing firm evidence that SACs may be more durable than nanocluster/NP catalysts under working conditions. In conclusion, tuning metal–support interaction is an effective strategy for designing superior catalysts for various reactions.

## 5. Summary

In conclusion, much progress has been made in the field of single-atom catalysis, especially for the rational design of novel catalysts and continuous expansion of reaction scopes. However, we should note that there are still numerous gaps between theoretical calculations and experimental results, so it is necessary to conduct advanced modeling design, parameter optimizing, and arithmetic system to improve theoretical calculations. Characterization to identify the singly dispersed species on the support has been developed as a reliable method combing electron microscopic imaging, infrared and electronic spectrum analysis. But the real structure or component of active sites during catalytic process are still concealed in the "black box". In addition, a quantitative explanation of interactions between active metal and neighboring atoms on the support is almost a blank, which hampers the thorough understanding of the true nature in single-atom catalysts.

**Author Contributions:** Y.G. wrote the first draft of the article which was then modified by R.L. and B.Q.

**Funding:** This work was supported by National Natural Science Foundation of China (21606222, 21776270), Strategic Priority Research Program of the Chinese Academy of Sciences (XDB17020100), the National Key R&D Program of China (2016YFA0202801).

**Conflicts of Interest:** The authors declare no conflict of interest.

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
