# Peer review of "Highlights of Major Progress on Single-Atom Catalysis in 2017"

_catalysts, doi:10.3390/catal9020135_

Round 1

Reviewer 1 Report

In the manuscript named highlights of the major progress on the single-atom catalysis in 2017, their authors show in a format of mini-review the major achievements and progress in the single-atom catalysis during the year 2017. This mini-review is well written and accessible to a broad readership. However, there are some suggestions that maybe can help to improve the quality of the manuscript. Before recommending publications, I encourage the authors to will take the following minor and major suggestions into account.

Suggestions and comments are attached here:

# Minor

1.      In comparison with another figure shown in the manuscript, figure 3 is briefly and poorly explained. The authors must explain it better.

2.      Line 174, it appears in the text “... methane is a holy-grail in modern… “ Replace “holy-grail” for another term nonreligious. Scientific vocabulary is quite rich, authors must avoid religious terms

3.      Line 187, reference [40] is not well located, I guess that should be before of the dot.

4.      In section 3.5 and 3.6, there are some typos in the references.

a.             For example, [68], [69] should be [68,69].

b.             For example, [65], [66] should be [65,66].

c.             For example, [71], [72] should be [71,72].

d.            For example, [73], [74] should be [73,74].

5.      Line 273, typo in hydrosilation[75], should be hydrosilation_[75].

6.      Line 296, the authors show here a conclusion. I recommend them to use this conclusion in the summary part.

7.      Line 306, the adjective "drastic " is not well used by the authors. They can use adjectives as extreme or radical.

8.       Line 312, Authors said that there are numerous gaps between theoretical calculations and real working conditions. I recommend them to replace the part of “real working conditions” by ”experimental results”.  

9.      Moderate the conclusion that only theoretical calculations require to be improved. Are the experiments well designed? Aren´t there any limitations in the experiments? Are the experiments perfect? Or maybe both parts should be improved?

# Major

1.      The Scanning tunneling microscope (STM) and the electrochemical-STM experiments are contributing to the understanding of the heterogeneous catalysis. In fact, STM is the unique technique that can move and control the atoms on the surface, opening the door for the understanding catalysis at single atom level. It was quite surprising, that in this manuscript there aren’t any paragraph, sentence or subsection that talk about these two approaches and their achievements, despite that in 2017 it was published papers as, “Direct instrumental identification of catalytically active surface sites. Jonas H. K. Pfisterer,  Yunchang Liang,  Oliver Schneider, Aliaksandr S. Bandarenka Nature volume 549, pages 74–77 (07 September 2017)”. So, the authors must include a paragraph or a subsection that show the advances and achievements of these two approaches, including references.

Author Response

Point 1 (Minor): In comparison with another figure shown in the manuscript, figure 3 is briefly and poorly explained. The authors must explain it better.

Response Minor 1: Thanks for your suggestion. I have revised this part, and in the new revision it is explained as following: Wang et al. compared traditional UV irradiation with UV irradiation accompanying iced-photochemical process. Obviously, the traditional UV irradiation of H2PtCl6 aqueous solutions (as shown in the upper line) produced Pt nanocrystals formed by the agglomeration and nucleation of Pt atoms. However, Pt single atoms dispersed in ice could be attained by exposing the frozen solution of H2PtCl6 to UV irradiation with a low temperature to reduce Pt4+. However, there is something wrong with the copyright permission of this figure, so we have deleted this one.

Point 2 (Minor): Line 174, it appears in the text “methane is a holy-grail in modern…” Replace “holy-grail” for another term nonreligious. Scientific vocabulary is quite rich, authors must avoid religious terms

Response Minor 2: Thanks for your suggestion. I  am sorry to use the inappropriate expression, and I have changed the word “holy-grail” to the phase “challenging process of great importance”.

Point 3-5 (Minor):  3. Line 187, reference [40] is not well located, I guess that should be before of the dot; 4. In section 3.5 and 3.6, there are some typos in the references. a. For example, [68], [69] should be [68,69]. b. For example, [65], [66] should be [65,66]. c. For example, [71], [72] should be [71,72]. d. For example, [73], [74] should be [73,74]; 5. Line 273, typo in hydrosilation[75], should be hydrosilation [75].

Response Minor 3-5: Thanks for you pointing out most of the careless errors of the references. I have revised all of those in the new revision.

Point 6 (Minor):  Line 296, the authors show here a conclusion. I recommend them to use this conclusion in the summary part.

Response Minor 6: Thanks for your suggestion. I have used this conclusion in the last of part 4 to summarize.

Point 7 (Minor): Line 306, the adjective "drastic " is not well used by the authors. They can use adjectives as extreme or radical.

Response Minor 7: Thanks for your suggestion. I  am sorry to use the inappropriate expression,  and I have changed the word “drastic” to “extreme”.

Point 8 (Minor):  Line 312, Authors said that there are numerous gaps between theoretical calculations and real working conditions. I recommend them to replace the part of “real working conditions” by “experimental results”.

Response Minor 8: Thanks for your suggestion. I  am sorry to use the inappropriate expression, and I have changed the phase “real working conditions” to “experimental results”.

Point 9 (Minor): Moderate the conclusion that only theoretical calculations require to be improved. Are the experiments well designed? Aren´t there any limitations in the experiments? Are the experiments perfect? Or maybe both parts should be improved?

Response Minor 9: I  am sorry to make a confusion, and I have revised to “ However, we should notice that there are still numerous gaps between theoretical calculations and experimental results, so it is necessary to conduct advanced modeling design, parameter optimizing, and arithmetic system to improve theoretical calculations.”

Point 1 (Major): The Scanning tunneling microscope (STM) and the electrochemical-STM experiments are contributing to the understanding of the heterogeneous catalysis. In fact, STM is the unique technique that can move and control the atoms on the surface, opening the door for the understanding catalysis at single atom level. It was quite surprising, that in this manuscript there aren’t any paragraph, sentence or subsection that talk about these two approaches and their achievements, despite that in 2017 it was published papers as, “Direct instrumental identification of catalytically active surface sites. Jonas H. K. Pfisterer, Yunchang Liang, Oliver Schneider, Aliaksandr S. Bandarenka Nature volume 549, pages 74–77 (07 September 2017)”. So, the authors must include a paragraph or a subsection that show the advances and achievements of these two approaches, including references.

Response Major 1: Thanks for your suggestion. In this mini-review, since we summarize the major achievements of single-atom catalysts (SACs) in 2017 into three main aspects: a) the advanced synthetic methodologies, b) catalytic performance in C1 chemistry, c) strong metal-support interaction induced unexpected durability, there is no specific part on advanced characterization techniques for SACs. The Scanning Tunneling Microscope (STM) and the electrochemical-STM experiments are contributing to the understanding of the heterogeneous catalysis and STM is the unique technique that can move and control the atoms on the surface opening the door for the understanding catalysis at single atom level. Since the characterization technique is vital for the development of heterogeneous catalysis and SACs, we have cited the paper to make our minireview more complete in the introduction of the new revision with the reference [11] (Nature 2017, 549, 74–77). 

Reviewer 2 Report

The authors have presented a mini-review manuscript entitled ‘Highlights of the major progress on single-atom catalysis in 2017”. As represented by the title, the authors summarized the progress and significant discoveries in the emerging field of Single-Atom Catalysis (SAC) published in 2017.  The authors have diligently drafted the manuscript by covering the three important aspects such as 1. Synthetic methodologies of SACs, 2. Applicaions of SACs in conversion of C1 chemicals (main constituents of natural gas and fossil chemicals) into the industrially important chemicals, which have the potential implications to meet the future energy needs. 3. Strong metal-support interaction of SACs leading to unexpected stability.

Considering the emerging applications of SACs, these kinds of review articles are important (useful) to the community working in this field.

Minor point: Authors should also include the citation for the following latest review article.

1. Nature Reviews Chemistry, 2, pages 65–81, 2018.            

Author Response

Point 1 (Minor point): Authors should also include the citation for the following latest review article. 1. Nature Reviews Chemistry, 2, pages 65–81, 2018.            

Response 1: Thanks for your suggestion. The latest review article (Nature Reviews Chemistry 2018, 2, 65–81.) have summarized experimental and computational efforts aimed at understanding the bonding in single-atom catalysts and how this related to catalytic performance, which was of enlightened importance  for the understanding of heterogeneous catalysis at single atom level. Thus, we cited that review to make our minireview more complete in the introduction with the reference [10].

Reviewer 3 Report

Reviewer’s comment

 Manuscript number: Catalyst-

 Manuscript title: Highlights of the major progress on single-atom catalysis in 2017

 Manuscript authors: Yalin Guoa, Rui Langa, Botao Qiao

The authors presented a review of articles published during the period of 2012-2017 (mainly 2017) in a topic entitled as “Single-Atom Catalysis”. The authors attempted to consider the latest achievements in the fields of both preparation technique of such materials and their application in new catalytic processes.

Unfortunately, these fields were not coordinated between each other in the review, because the References consisted of the results of application of the single-atom catalysts in the C1-conversion processes were not analyzed in terms of preparation technique. In addition, a brief description of the references in the field, for example, “CH4 conversion” subsection, does not provide the opportunity to compare the traditional catalysts used for partial methane oxidation and the novel single-atom catalysts. No analysis of the advantages and disadvantages of such novel single-atom catalysts is presented in the manuscript.

The structure of the manuscript requires changing. For instance, the volume of the main four parts of the review, including parts 2-4, is not equal. So, if the third part consists of six subsections, the forth part occupies just one half of the page and includes only a brief description of 6 references. Moreover, I think that the title of the section “Strong metal-support interaction of SACs” is not suitable for the discussed topic, since initially the existence of the single-atom active sites is a result of the isolation of atoms from each other on surface (and/or bulk) due to the energy exchange between, for example, functional groups of the supports and single atoms, etc. I suggest including the forth part into the discussion of new approaches in the preparation techniques (at the end of the second part).

I want to underline the carelessly designed reference list. This list consists many different errors, mistakes, etc.  (details in file).

Author Response

Point 1: Unfortunately, these fields were not coordinated between each other in the review, because the References consisted of the results of application of the single-atom catalysts in the C1-conversion processes were not analyzed in terms of preparation technique. In addition, a brief description of the references in the field, for example, “CH4 conversion” subsection, does not provide the opportunity to compare the traditional catalysts used for partial methane oxidation and the novel single-atom catalysts. No analysis of the advantages and disadvantages of such novel single-atom catalysts is presented in the manuscript.

Response 1: Thanks for your nice comments, following are our response.

(1) Part 2 emphasizes some new preparation strategies of stable single-atom catalysts (SACs), whereas Part 3 emphasizes the application of SACs especially in some important C1 reactions. It is noteworthy the novel preparation method of SACs applied to important C1 conversion is still mentioned if necessary, such as in section 3.1. But most SACs applied to C1 conversion are prepared by the traditional methods, so their preparation details are not involved in this review. The connection between these two parts is that they are both beneficial to understand the nature of SACs and the two parts both are meaningful to the future research of SACs. Of course, we hope in the future, some new preparation strategies may be adopted to SACs, and these SACs may be used in C1 conversion and some other reactions.

(2) In this review, the catalytic performance of SACs and traditional nanoparticles (NPs) is not compared one by one, because the experimental conditions (working temperature, pressure, catalyst usage etc.) are different in every report. In addition, as mentioned in the introduction, many previous reviews have illuminated the differences between SACs and NPs. Therefore, we did not talk a lot on this point, but we did give some examples in the text, e.g. the 1st paragraph of Part 2 (Au SACs vs Au NPs), the section 3.1 (Rh/ZrO2 SACs vs Rh NPs) and so on. The advantages and disadvantages of SACs are discussed in almost every reports and reviews, such as the maximum utilization of atoms, high selectivity, and so on. Thus, we would like to focus on the new achievements in this field, avoiding such comments.

Point 2: The structure of the manuscript requires changing. For instance, the volume of the main four parts of the review, including parts 2-4, is not equal. So, if the third part consists of six subsections, the forth part occupies just one half of the page and includes only a brief description of 6 references. Moreover, I think that the title of the section “Strong metal-support interaction of SACs” is not suitable for the discussed topic, since initially the existence of the single-atom active sites is a result of the isolation of atoms from each other on surface (and/or bulk) due to the energy exchange between, for example, functional groups of the supports and single atoms, etc. I suggest including the forth part into the discussion of new approaches in the preparation techniques (at the end of the second part).

Response 2: Thanks for your sincere advice to add the forth part into the discussion of the preparation techniques. We totally agree with you that the isolated metal atoms are preserved by some unique properties (defects, vacancies, heteroatom-containing groups etc.) of the supports, and we summarized these contents in Part 2. However, in Part 4, we would like to highlight that a deeper understanding about the strong metal-support interaction (SMSI) would help us to tune the catalytic activity and durability of SACs. During 2017, SMSI of SACs was mentioned in a few reports, but from then on, more efforts were devoted in this field. For example, recently Lang et al. (nature communication 2019, 10, 234.) reported that isolated Pt atoms can be stabilized through a strong covalent metal-support interaction (CMSI) paving a new way for constructing high-loading SACs for diverse industrially important catalytic reactions. Thus, we prefer to emphasize the SMSI of SACs in a separate part.

Point 3: I want to underline the carelessly designed reference list. This list consists many different errors, mistakes, etc.  (details in file).

Response 3: Thanks for you reading my review carefully and pointing out most of the careless errors of the references. I  am sorry to use many inappropriate expressions, and I have revised all of them and some other mistakes in the new revision.

Round 2

Reviewer 1 Report

In this new version of the manuscript named Highlights of the major progress on the single-atom catalysis in 2017, their authors show an improved manuscript. Under my point of view. I recommend this new version to be published.

Sincerely

Reviewer 3 Report

Authors corrected significantly the manuscript.

I think that revised version of the manuscript may be accepted in the present form.